# Review: Targeting the Transforming Growth Factor-Beta Pathway in Ovarian Cancer

**DOI:** 10.3390/cancers11050668

**Published:** 2019-05-14

**Authors:** Brandon M. Roane, Rebecca C. Arend, Michael J. Birrer

**Affiliations:** 1Department of Obstetrics and Gynecology-Gynecologic Oncology, University of Alabama at Birmingham, Birmingham, AL 35233, USA; rarend@uabmc.edu; 2O’Neal Comprehensive Cancer Center, University of Alabama at Birmingham, Birmingham, AL 35233, USA; mbirrer@uab.edu

**Keywords:** ovarian cancer, transforming growth factor-β, TGF-β, biomarkers, proliferation

## Abstract

Despite extensive efforts, there has been limited progress in optimizing treatment of ovarian cancer patients. The vast majority of patients experience recurrence within a few years despite a high response rate to upfront therapy. The minimal improvement in overall survival of ovarian cancer patients in recent decades has directed research towards identifying specific biomarkers that serve both as prognostic factors and targets for therapy. Transforming Growth Factor-β (TGF-β) is a superfamily of proteins that have been well studied and implicated in a wide variety of cellular processes, both in normal physiologic development and malignant cellular growth. Hypersignaling via the TGF-β pathway is associated with increased tumor dissemination through various processes including immune evasion, promotion of angiogenesis, and increased epithelial to mesenchymal transformation. This pathway has been studied in various malignancies, including ovarian cancer. As targeted therapy has become increasingly prominent in drug development and clinical research, biomarkers such as TGF-β are being studied to improve outcomes in the ovarian cancer patient population. This review article discusses the role of TGF-β in ovarian cancer progression, the mechanisms of TGF-β signaling, and the targeted therapies aimed at the TGF-β pathway that are currently being studied.

## 1. Introduction

Ovarian cancer is the fifth most common malignancy among women in the United States and is the leading cause of gynecologic cancer-related deaths. In 2018, over 22,000 women were diagnosed with ovarian cancer, and 14,000 women die from this disease [1]. Due to the lack of effective screening strategies and the non-specific symptoms prior to presentation, the vast majority of women with ovarian cancer present with advanced-stage disease (International Federation of Gynecology and Obstetrics (FIGO) stage III or IV) [2,3]. The standard initial therapy for advanced-stage ovarian cancer is primary debulking surgery followed by adjuvant chemotherapy or neoadjuvant chemotherapy followed by interval debulking and additional chemotherapy. Upfront therapy results in disease remission in approximately 75% of patients [2,3,4]. Despite the success of primary treatment in ovarian cancer, almost all women with advanced disease recur, resulting in a five-year survival of 30%. Once a patient experiences a recurrence, this is often characterized as a chemoresistant disease and the response rate to second-line agents is very low. A major challenge in developing new and effective treatment options for ovarian cancer is the heterogeneous nature of the disease. Even within patients diagnosed with high-grade serous epithelial ovarian cancer (EOC), by far the most common histology of ovarian cancer, there is a wide range of clinical outcomes and responses to therapy [5]. Current research is aimed at discovering biomarkers that are both prognostic and potential targets for novel therapies [6]. Targeted therapy allows for an individualized approach in the management of ovarian cancer based on the patient’s genetics and the molecular characteristic of the tumor. Understanding the underlying molecular biology of ovarian cancer is key to developing targeted therapies that can be used in addition to standard treatment modalities to improve outcomes in these patients.

The transforming growth factor-β (TGF-β) superfamily is a class of over 30 proteins including TGF-β isoforms 1, 2, and 3, activins, bone morphogenetic proteins (BMPs), growth factors, differentiating factors, and anti-Müellerian hormone (AMH) [7,8,9]. Signaling via the TGF-β pathways is used to modulate a wide variety of cellular processes including cell growth and differentiation, apoptosis, adhesion, invasion, angiogenesis, and immune regulation [7]. The TGF-β superfamily plays an important role in embryogenesis and tissue homeostasis in human adults [10,11]. This diversity of function is accomplished through a complex network of intracellular signaling but is also influenced by positive and negative feedback controlled by extracellular cues [12]. Dysregulation of this pathway plays an important role in tumorigenesis and progression [13]. Focusing on its role in cancer development, TGF-β displays a dichotomous behavior as both a tumor suppressor and a promoter [14,15]. In early-stage cancers, TGF-β functions to suppress cell growth and promote apoptosis, and loss of TGF-β function is associated with tumor development [16,17]. However, in advanced-stage malignancies, overexpression of TGF-β has been linked to a more aggressive and metastatic phenotype [18]. This “TGF-β Paradox” has been the focus of recent research in order to better understand its oncogenic activity [19].

The TGF-β superfamily has demonstrated the same general breadth of activity within the normal ovary and in ovarian cancer as well. Members of the TGF-β family of proteins are involved in signaling between the cells of the normal ovary leading to follicle development [19]. The duality of TGF-β as both a tumor suppressor and promoter of progression have been documented in ovarian cancer as well [20,21,22]. In particular, TGF-β has been implicated in creating an environment where ovarian cancer cells are able to evade the host immune system defenses aimed at controlling tumor growth. This allows for increased tumor dissemination and worse prognosis in ovarian cancer patients [23,24]. Given its role in tumor progression, there has been interest in developing targeted therapies against TGF-β in cancer patients [25]. Therapies aimed at inhibiting the tumor-promoting activity of TGF-β have been demonstrated in a variety of cancer types including breast, prostate, pancreas, and ovarian cancer [26]. This review summarizes TGF-β signaling and research focused on understanding the role of TGF-β in patients with advanced ovarian cancer, as well as targeted TGF-β therapy being studied to improve outcomes in this patient population.

## 2. The TGF-β Signaling Pathway

There are three distinct TGF-β isoforms, TGFβ1, TGFβ2, and TGFβ3, each with unique functions in vivo [27]. TGF-β is produced by a variety of cell types as an inactive molecule bound to the latency-associated protein (LAP). This forms a complex with another protein—latent-TGF-β binding protein (LTBP)—which guides the inactivated TGF-β complex towards the cell membrane where it undergoes activation via the interaction with proteases within the extracellular matrix [28,29,30]. Activated TGF-β then binds to the transmembrane protein—transforming growth factor-β receptor-2 (TβRII)—which recruits the transforming growth factor-β receptor-1 (TβRI). In the canonical pathway of TGF-β signaling, TβRI utilizes serine/threonine kinase activity to conduct intracellular signaling via phosphorylation of cytoplasmic proteins Smad2 and Smad3. Smad2 and Smad3 combine with Smad4 and form a heteromeric complex which is able to translocate to the nucleus [31]. The nuclear Smad complex binds to TGF-β effector genes for transcriptional regulation [32].

The same transmembrane signaling through TβRI and TβRII could conduct TGF-β signaling in a Smad-independent manner. The TGF-β non-canonical pathway triggers well-known intracellular pathways including mitogen-activating protein kinases (MAPK), Rho-like GTPase, phosphatidylinositol-3-kinase/AKT, and JNK/p38. Together, the TGF-β canonical and non-canonical pathways regulate a wide array of functions along with a combination of various mechanisms used to control TGF-β secretion and activation, the post-translational modifications of Smad proteins, and cell-specific co-factors governing binding to the target genes (Figure 1) [6,33,34,35].

## 3. TGF-β Signaling in Ovarian Cancer

Dysregulation in TGF-β signaling has been shown to be associated with ovarian carcinogenesis. TGF-β is a potent inhibitor of cellular growth and loss of function in the TGF-β pathway can result in uncontrolled proliferation leading to tumor development [13,36,37]. In the normal ovary, TGF-β activity suppresses cellular growth and promotes differentiation. In vitro studies have shown that TGF-β inhibits the growth of ovarian cancer cell lines [38,39]. However, in 40% of ovarian cancers, there is a loss of this cytostatic function [40]. At the receptor level, resistance to TGF-β signaling is a mechanism in the establishment of ovarian cancer. Mutations in both TβRI and TβRII have been found in a variety of cancer types including ovarian cancer. In a study by Chen et al., approximately half of their primary ovarian cancer samples analyzed by PCR possessed a significant variant in the coding region of the TβRI gene [41]. Similarly, 50% of ovarian cancer samples analyzed by Lynch et al. had a loss or decreased expression of TβRII [42]. However, there are conflicting reports showing that mutations in TGF-β receptor genes are an infrequent occurrence in ovarian cancer, indicating that there are potentially additional mechanisms at play in the role of the TGF-β pathway in this disease [43,44,45].

While mutations in Smad gene sequences are uncommon [46,47], the reduced expression and decreased binding ability of the Smad4 protein, which is required for the nuclear translocation of TGF-β signaling through the canonical pathway, is found at a higher frequency in ovarian cancers compared to normal ovarian tissue [48,49]. The loss of Smad4 confers a more aggressive phenotype as the tumor suppressive function of TGF-β is lost, but epithelial-to-mesenchymal transitioning (EMT) activity is preserved [50]. In comparison, in the study by Dunfield et al., Smad-dependent TGF-β signaling is shown to be intact in primary human ovarian cancer cell lines [51]. Further, Hurteau et al. showed that canonical signaling is preserved in ovarian cancer ascites cell lines [52].

Ovarian cancer is classically divided into three different tumor types: epithelial, sex-cord stroma, and germ cell tumors, with over 90% classified as epithelial malignancies. Within the epithelial tumors alone, there are five main histotypes: low grade and high grade serous, clear cell, endometrioid, and mucinous carcinomas [53]. At a histological level, great heterogeneity exists which is further characterized on a molecular level. Despite these well-documented differences, which greatly impact prognosis and clinical outcomes, similar treatment strategies and therapies are utilized across all ovarian cancer subtypes. It is important to identify pathways with differential signaling that contribute to the development of these malignancies. Existing literature supports the hypothesis that disruption of TGF-β superfamily genes results in the development of human sex-cord stromal tumors, especially granulosa cell tumors (GCT) [54]. New evidence has revealed molecular and functional interactions between Smad3, a member of the Smad complex involved in the intracellular signaling of the TGF-β canonical pathway, and FOXL2 [55,56]. This interaction regulates the activity of genes associated with cell proliferation and survival suggesting that Smad3 activation could serve as a critical converging point between the dysregulated TGF-β superfamily signaling and the genetic alterations in human GCT development. Indeed, FOXL2 is reported to be mutated in most granulosa cell tumors [57]. In this regard, changes in FOXL2, particularly hypermethylation, have been reported recently in ovarian carcinomas especially in serous and mucinous histotypes and seem to be associated with a poorer prognosis [58]. Therefore, the interaction between FOXL2 alterations and the TGF-β superfamily signaling pathway will help understand the pathogenesis of ovarian tumors and design novel therapeutic strategies, especially in this particular histological subtype.

Looking further downstream in the TGF-β pathway, regulatory genes associated with the biological inhibition of TGF-β have been identified and shown to be altered in ovarian cancer. Genes such as DACH1 and EV11 are overexpressed in ovarian cancer and block the anti-proliferative effects of TGF-β. These genes produce proteins that interact with the Smad2/3 complex suppressing its transcriptional activity [59,60]. This effect is more prominent in the early stage than in advanced or recurrent ovarian cancer [61]. Nuclear Smad interacts with a variety of transcriptional corepressors including two proto-oncogenes Ski and SnoN. TGF-β stimulation results in the dissolution of this association allowing for signal transduction [62]. When comparing normal ovarian epithelium to ovarian carcinoma, TGF-β resistance does not produce alterations in the expression or degradation of these two corepressor genes [22]. Another downstream target of Smad proteins includes c-Myc, where TGF-β signaling results in down-regulation of c-Myc, promoting cell cycle arrest [63]. Lack of response to TGF-β could be explained by the uncontrolled expression of c-Myc which coincides with the resistance to the antimitogenic activity of TGF-β in ovarian cancer [22,64]. Smad-independent aberrations have also been identified as drivers of tumorgenesis. There is evidence that within cancer cells, TβRI undergoes cleavage by the tumor necrosis factor (TNF) alpha converting enzyme (TACE) resulting in the translocation of the receptor’s intracellular domain to the nucleus. This leads to activation of the p38 mitogen-activated kinase (MAPK) pathway which has been associated with EMT [65]. Further, ubiquitin-specific protease (USP) 4 has also been implicated in the cleavage of TβRI. This cleavage activity of the transmembrane receptor via USP4 controls the receptor levels along the cellular membrane. Depletion of USP4 blocks TGF-β-induced EMT. The stability of USP4 is dependent on AKT phosphorylation which leads to cytoplasmic migration of USP4 from the nucleus to the cytoplasm and cellular membrane where it can act on TβRI. USP4 serves as a link between TGF-β and AKT signaling via cleavage of TβRI [66].

Loss of sensitivity to the tumor suppressive effects of TGF-β plays a critical role in tumorgenesis in ovarian cancer. However, TGF-β has an opposing functionality as a promoter of tumor progression at advanced stages. Once the cancer cells escape the inhibitory effect of TGF-β, they begin to overexpress TGF-β to create a more aggressive tumor phenotype through the promotion of EMT, increased tumor invasion, metastasis, and angiogenesis, as well as immune evasion [28].

Stimulation of the human ovarian cancer cell line SKOV-3 with TGF-β leads to morphological mesenchymal changes via the reorganization of the tumor cell cytoskeleton [67]. In advanced stages of ovarian cancer, TGF-β is prominent within the tumor microenvironment. TGF-β signaling triggers surrounding stromal cells promoting epithelial cell growth [68]. Yeung et al. demonstrated that ovarian cancer invasion is regulated by cancer-associated fibroblasts which have increased expression of TGF-β receptors and increased TGF-β signaling [69]. In a meta-analysis of the genomic profiles from 1525 ovarian cancer samples, overexpression of TβRII was part of a highly specific gene signature differentiating suboptimal versus optimally debulked patients undergoing primary cytoreductive surgery. As expected, the gene signature ability to predict suboptimal debulking surgery correlated with worse prognosis in these patients. Furthermore, out of the five overexpressed genes in the debulking gene signature, TβRII had the greatest statistical difference (*p* = 0.005). This finding was confirmed via immunohistochemistry (IHC) with suboptimally debulked patients having increased levels of phosphorylated Smad2/3 staining compared to optimally debulked patient samples [70]. The association between the debulking status and increased TGF-β signaling supports the premise that TGF-β promotes increased tumor dissemination leading to worse clinical outcomes in ovarian cancer patients.

As a primarily peritoneal disease, advanced-stage ovarian cancer frequently presents with the accumulation of ascites fluid due to increased production from tumor implants [71]. The conventional thought is that the presence of ascites fluid is linked to aberrations in the expression of vascular endothelial growth factor (VEGF) caused by the tumor burden [72]. However, in a study by Liao et al., TGF-β was important not only for the formation of ascites fluid [73], but was crucial in reducing ascites fluid drainage. In murine models, blockade of TGF-β resulted in decreased ascites formation via inhibition of VEGF and increased ascites drainage by preventing changes to the lymphatic channels resulting in the control of malignant ascites [74].

The cellular context and timing of TGF-β signaling is integral to its impact in tumorgenesis and progression. In ovarian cancer, which is a disease known to present in advanced stages, the impact of TGF-β as a promoter of tumor progression appears to dominate, making it an intriguing biomarker of ovarian cancer from both a prognostic and therapeutic standpoint.

## 4. TGF-β Regulated Immune Evasion

In general, host immunity has the capability of controlling tumor development as tumor antigens are recognized by host defenses which are then marked for elimination [75]. T-cells are a major component of this response and work together with antigen-presenting cells to activate humoral and cellular anti-tumoral immune responses. Cytokines have the ability to enhance or attenuate this response [76]. Eventually, the tumor microenvironment is altered, and immune surveillance fails. The tumor utilizes multiple strategies to evade host immunity and is allowed to grow, effectively undeterred by the immune system [75]. One specific strategy employed by tumor cells is the recruitment and production of tumor suppressive factors to help create an environment conducive to tumor growth and metastasis [76].

TGF-β is involved in regulating a variety of immune cell lines, including both innate and adaptive immune responses [77]. Under normal conditions, TGF-β regulates immune cell growth in an effort to maintain self-tolerance [28,76]. In malignancies, including ovarian cancer, TGF-β is a potent immunosuppressor within the tumor microenvironment, affecting natural killer and dendritic cell activity, cytokine production, and T-cell function [76,78,79,80,81]. One important mechanism to suppress cytotoxic T-cell function is through the increased production of regulatory T-cells (Tregs) [78]. Increased secretion of TGF-β within the tumor microenvironment recruits Tregs via expression of FoxP3, which ultimately results in diminished cytotoxic T-lymphocytes [82]. In ovarian cancer, Tregs have been shown to be associated with increased tumor growth and reduced survival [83]. In a study conducted by Wolf et al., elevated FoxP3 expression was an independent prognostic factor in 99 ovarian cancer samples. Increased FoxP3 expression was a marker of the presence of Tregs within the tumor, and this was associated with both a decrease in progression-free and overall survival [84]. In primary debulking surgeries, increased cytotoxic T-cell function and reduced Tregs is associated with improved rates of optimal debulking [85]. The ratio of tumor infiltrating lymphocytes (TILs) to Tregs was further explored by Leffers et al., who showed that this was an independent predictor of clinical outcomes in ovarian cancer patients, with an increase in TILs to Tregs associated with improved outcomes [86].

Immunosuppression via TGF-β signaling is a prominent factor in the progression of multiple cancers including ovarian cancer. The ability of the tumor to increase TGF-β levels and enhance TGF-β signaling facilitates tumor growth and dissemination by weakening cytotoxic immune defenses. The potential for immune therapy to abrogate the effect of TGF-β in the advanced ovarian cancer setting is a current area of research and is further summarized below.

## 5. Targeting TGF-β in Ovarian Cancer

Anti-TGF-β therapy has been studied extensively in a wide variety of tumors via in vitro studies, preclinical models, and clinical trials. The ability of TGF-β to create an environment to promote tumor growth presents potential opportunities for therapeutic options to control disease progression. Numerous anti-TGF-β agents have been studied in different tumor types, and some have made their way to clinical trials. Targeted therapies have been designed to affect TGF-β signaling at a variety of points along the pathway. The most common mechanisms implemented include (1) preventing TGF-β synthesis using antisense molecules; (2) inhibition of TGF-β binding at the receptor level; (3) blocking kinase activity which eliminates the intracellular signaling through Smad signaling proteins; and (4) immune-based response strategies [28,87]. The vast majority of women with ovarian cancer will develop resistance to conventional cytotoxic therapies used in the upfront setting. Recent advances in research have led to a better understanding of the molecular characteristics of ovarian cancer and the development of new targeted therapies [88].

Immunotherapy is currently being studied as a potential method for the control of ovarian cancer progression. The utilization of immune checkpoint inhibitors has gained significant momentum over the past several years as they have demonstrated success in solid tumors [89]. In particular, therapy aimed at the programmed cell death 1 (PD-1) and the cytotoxic T-lymphocyte associated antigen 4 (CTLA-4) pathways are currently being studied in numerous clinical trials in ovarian cancer [90,91]. These studies have demonstrated success in a subset of patients with ovarian cancer, and there has been a shift towards understanding biomarkers that correlate with a response as well as options for additional targets to work in combination with these therapies [92]. Tauriello et al. designed a quadruple-mutant mouse model that spontaneously generates metastatic intestinal tumors. Anti-PD-L1 therapy yielded a limited response in the model; however, inhibition of TGF-β with a small molecule inhibitor produced a robust cytotoxic T-cell response preventing metastasis and increasing susceptibility to anti-PD-L1 therapy [93]. The combination of anti-TGF-β therapy and immune checkpoint inhibition has also been studied in a preclinical breast cancer model. In vivo treatment of mice with a TGF-β inhibitor resulted in almost 100% tumor growth inhibition with regression in about half the mice once therapy was discontinued. In those mice that had a durable regression, upon re-challenge with 4T1-LP tumor cells, they remained tumor-free suggesting the establishment of immunological memory. This same study combined anti-PD-L1 antibody with anti-TGF-β therapy in a colon cancer mouse model which resulted in an improved response when compared to anti-PD-L1 alone [94]. In a study by Mariathasan et al., increased levels of TGF-β correlated with a limited response to anti-PD-L1 therapy and that blockade of TGF-β signaling along with administration of anti-PD-L1 therapy resulted in an even greater reduction of TGF-β signaling, increased the penetration of infiltrating T-cells, and a reduction of tumor growth [95]. A pair of studies described fusion proteins that simultaneously target TGF-β receptors and PD-L1 along with TGF-β receptors and CTLA-4. This combined blockade strategy, resulted in a marked reduction of tumor burden in preclinical models, further supporting the growing body of literature that supports the role of TGF-β inhibition in combination with immunomodulatory agents [96,97]. These combination strategies are not limited to immune checkpoint inhibitors as outlined above, but also include OX40 antagonistic antibodies [98], IL-2 [99], radiation therapy [100], and oncolytic viruses [101]. The results from these studies show early promise for increased efficacy of immunotherapy and are potentially applicable in the ovarian cancer patient population.

There have been increased efforts investigating anti-TGF-β therapy in ovarian cancer specifically. Trabedersen is a synthetic TGF-β2 antisense oligonucleotide therapy which has been tested in brain, pancreatic, prostate, and colorectal cancers [26]. Phase II data showed that when compared to conventional chemotherapy, patients with glioblastoma treated with Trabedersen had an overall survival of 39.1 months versus 21.7 months, although this difference was not statistically significant [102]. In a phase I trial of advanced staged melanoma patients, Trabedersen was tested in combination with immune-based therapies as a way of increasing sensitivity to these agents. Increased overall survival was found, in particular in patients treated with subsequent chemotherapy [103]. Based on this data, preclinical ovarian cancer models were developed studying the tolerability and efficacy of Trabedersen in combination with paclitaxel. Subcutaneous human ovarian tumor xenografts using the SKOV-3 cell line were developed in mice. In one study, the combination of Trabedersen with paclitaxel was well tolerated as both a single agent and in combination, significantly reducing tumor burden, and increasing overall survival when compared to paclitaxel alone (*p* < 0.05) [104].

Small molecule inhibitors targeting kinase activity along the TGF-β signaling pathway are currently in use in different ovarian cancer studies. Protein kinase inhibition has been studied as a single therapy and in combination with standard therapy. LY2109761, a small molecule inhibitor of TβRI/II kinase activity, demonstrated an ability to reduce drug resistance specifically in ovarian cancer mouse models when used in combination with Cisplatin [105,106]. Galunisertib (LY2157299), an agent targeting the tyrosine kinase function of cell membrane TGF-β receptors, is being used in an ongoing clinical trial. Preliminary results show that this drug has the ability to amplify the effects of Nivolumab, a PD-L1 inhibitor, in a variety of solid tumors [107]. Furthermore, galunisertib has been tested in preclinical models and was found to attenuate the ovarian cancer cell line response to TGF-β signaling, as well as reduce tumor burden and ascites formation in ovarian cancer patient-derived xenograft (PDX) models [108]. A-83-01 specifically targets TβRI, and was used in the study of a syngeneic mouse model with the OV2944-HM-1 cancer cell line. This study showed a reduction in tumor dissemination by measuring peritoneal and omental implants [109]. Crosstalk between TGF-β and other well-known molecular pathways are being studied in ovarian cancer models as well. Inhibition of TGF-β signaling via LY2109761 was shown to reduce tumor burden in orthotopic ovarian cancer PDX models. This inhibition resulted in a reduction in mRNA expression of the insulin growth factor 1 (IGF-1) receptor. The therapeutic effect of TGF-β inhibition is lost when IGF-1 receptor signaling is blocked. These findings indicate that IGF-1 receptor signaling is positively regulated by TGF-β and is associated with ovarian tumor progression [22].

Vaccination strategies against TGF-β are in development and are being used to enhance the immune system response to tumor cells. Belagenpumatucel-L is an allogeneic whole cell vaccine constructed with a tumor-associated glycoprotein. The vaccine co-expresses anti-TGF-β antisense oligonucleotide and a vector for increased granulocyte macrophage colony-stimulating factor (GM-CSF. In a clinical trial studying patients with advanced melanoma, the vaccine was manufactured with a 84% success rate with 23 patients receiving at least one dose of the vaccination. No grade 3 or 4 toxicities were encountered and 35% of patients demonstrated a durable response in a disease that is particularly aggressive with dismal prognosis [110]. Comparable results were produced in a phase I study using the same anti-TGF-β shRNA/GM-CSF vaccine against patients with Ewing’s sarcoma [111]. Preliminary results from a phase II study showed that Vigil (Gemogenovatucel-T), a similarly designed bifunctional vaccine, used in conjunction with Atezolizumab had positive results in ovarian cancer patients. Twenty-nine patients with recurrent ovarian cancer have been enrolled to date. The median survival had not yet been reached at the time the data was reported, but out of the 29 patients, 20 had reached three years of survival, and the only reported adverse event was fatigue [112]. These results have led to the initiation of the VITAL Trial: Trial of Maintenance Vigil for High-Risk Stage IIIb–IV Ovarian Cancer. This is an active, phase II, randomized control trial where women with advanced stage ovarian, fallopian tube, or peritoneal carcinoma are randomly assigned to the administration of Vigil vaccination versus placebo following a complete response after upfront therapy. The primary outcome in this study is recurrence-free survival compared between the two arms [113].

While there is scarce data published regarding the utilization of anti-TGF-β monoclonal antibodies specifically in ovarian cancer, this therapy has been studied in various other tumor sites [114]. Newsted et al. identified a lead inhibitory antigen-binding fragment (Fab) via screening of a synthetic library with TβRII. Suppression with this lead Fab resulted in the reversal of the EMT and tumor invasion and an improved response to chemotherapy in EOC xenograft and syngeneic models [115]. Genzyme (1D11) is a neutralizing monoclonal antibody active against all three isoforms of TGF-β and was shown to reduce tumor dissemination in breast cancer [116,117,118]. Multiple preclinical trials have reported the effective use of receptor-neutralizing antibodies as well as soluble receptor molecules targeting TβRI and TβRII in different cancers [77,117,119]. Two ongoing trials studying GC1008, a humanized monoclonal antibody against TGF-β, are currently enrolling patients with mesothelioma and advanced melanoma in phase II trials [120,121]. These studies suggest that there is a future in studying TGF-β monoclonal antibodies in ovarian cancer cell lines, preclinical models, and eventually clinical trials.

## 6. Conclusion

The ability to exploit TGF-β signaling for potential therapeutic options has shown promise in a variety of malignancies including ovarian cancer as described above. The link between TGF-β signaling and ovarian cancer progression has repeatedly been demonstrated. Although the explanation of this association has not been entirely revealed, multiple mechanisms have been discovered explaining the role of TGF-β in promoting tumor growth. In ovarian cancer, this is particularly interesting because it is often diagnosed at an advanced stage and has a propensity to recur within 1–2 years following upfront treatment [2]. This information has led to the development of multiple therapies aimed at reducing TGF-β signaling and its impact on the tumor microenvironment. This strategy has proven safe and effective in the early stages of research in treating ovarian cancer. These drugs have entered into clinical trials with other malignancies and are now being considered for testing in ovarian cancer patients. The results from preclinical trials have provided convincing evidence of anti-TGF-β therapy’s ability to reduce tumor growth and dissemination, and the results from clinical trials are encouraging [26]. These results continue to push for further drug development and understanding of this pathway in ovarian cancer.

## Figures and Tables

**Figure 1 cancers-11-00668-f001:**
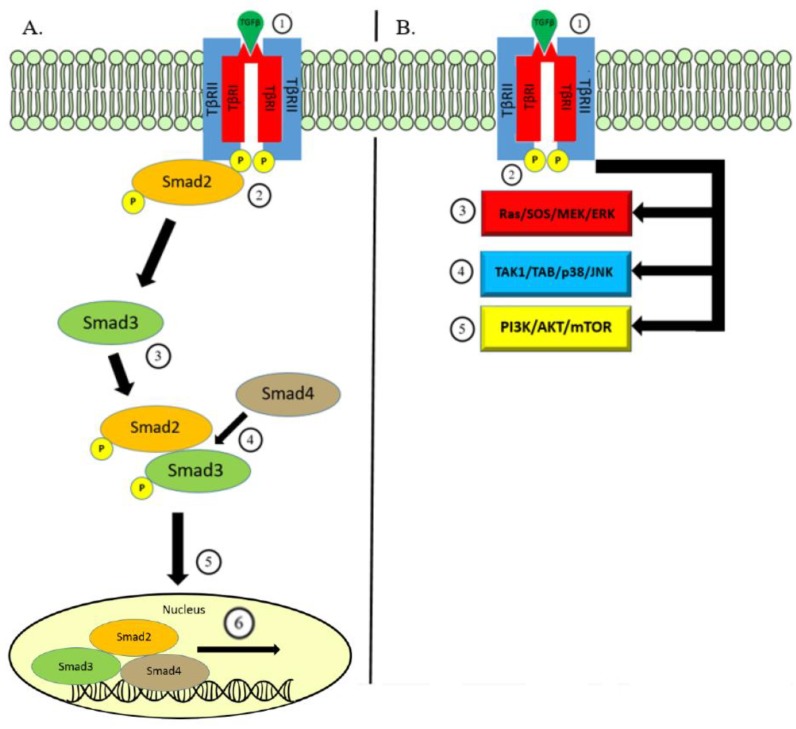
Overview of TGF-β Signaling. (**A**) Canonical Pathway: Signaling begins with binding of the TGF-β ligand on the TβRII which leads to the formation of a heteromeric complex with TβRI (1). This is translated to intracellular signaling through the kinase activity phosphorylating Smad2 protein (2). Smad3 is subsequently phosphorylated and binds to p-Smad2 (3). Together, phosphorylated Smad2/3 bind to Smad4 (4), and this protein complex is able to translocate (5) to within the nucleus and bind to genes responsive to TGF-β signaling to initiate transcription (6). (**B**) Non-Canonical Pathway: In a similar fashion, TGF-β ligand binding to cell surface receptors TβRI/II (1) can transmit signaling through the kinase activity (2) triggering other well-known pathways. Pathways involved in non-canonical TGF-β signaling include MAPK via the ERK (3) and JNK/p38 pathways (4) and the PI3K/AKT/mTOR pathway (5).

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
