# Peer review of "Review: Targeting the Transforming Growth Factor-Beta Pathway in Ovarian Cancer"

_cancers, 2019, doi:10.3390/cancers11050668_

Reviewer 1 Report

This review is well-structured and clear.

I have only few comments:

-          At page 2, I think it is  anti-Müllerian instead of Müellerian.

-          In the third section of the review, TGF-β signaling in ovarian cancer, I think that some informations could be added about the role of TGF-β pathway and its related genes in ovarian tumor in general, considering their great heterogeneity based on molecular, morphological and clinical characteristics, although epithelial  tumors are  the most frequent.

I suggest to introduce a paragraph, page 4, line 129: “Existing literature supports the hypothesis that disruption of TGF-β superfamily genes results in development of human sex-cord stromal tumors, especially  granulosa cell tumor (GCT) (Middlebrook BS, Eldin K, Li X, Shivasankaran S, Pangas SA. Smad1-Smad5 ovarian conditional knockout mice develop a disease profile similar to the juvenile form of human granulosa cell tumors. Endocrinology. 2009;150:5208-17; Mansouri-Attia N, Tripurani SK, Gokul N, Piard H, Anderson ML, Eldin K, Pangas SA. TGFβ signaling promotes juvenile granulosa cell tumorigenesis by suppressing apoptosis. Mol Endocrinol. 2014;28:1887-98). New evidence has revealed molecular and functional interactions between SMAD3, a member of Smad complex involved in the intracellular signaling of the TGF-β canonical patwhay, and FOXL2 (Anttonen M, Pihlajoki M, Andersson N, Georges A, L'hôte D, Vattulainen S, Färkkilä A, Unkila-Kallio L, Veitia RA, Heikinheimo M. FOXL2, GATA4, and SMAD3 co-operatively modulate gene expression, cell viability and apoptosis in ovarian granulosa cell tumor cells. PLoS One. 2014 Jan 9;9(1):e85545. doi: 10.1371/journal.pone.0085545. eCollection 2014; Nonis D, McTavish KJ, Shimasaki S. Essential but differential role of FOXL2wt and FOXL2C134W in GDF-9 stimulation of follistatin transcription in co-operation with Smad3 in the human granulosa cell line COV434.Mol Cell Endocrinol. 2013 Jun 15;372(1-2):42-8. doi: 10.1016/j.mce.2013.02.020). This interaction leads to regulate  the activity of genes associated to cell proliferation and survival suggesting  that  SMAD3 activation could serve as a critical converging point between dysregulated  TGF-β superfamily signaling and genetic alterations in human GCT development. Indeed, FOXL2 is reported mutated in most   granulosa cell tumors (Shah SP, Köbel M, Senz J, Morin RD, Clarke BA, Wiegand KC, Leung G, Zayed A, Mehl E, Kalloger SE, Sun M, Giuliany R, Yorida E, Jones S, Varhol R, Swenerton KD, Miller D, Clement PB, Crane C, Madore J, Provencher D, Leung P, DeFazio A, Khattra J, Turashvili G, Zhao Y, Zeng T, Glover JN, Vanderhyden B, Zhao C, Parkinson CA, Jimenez-Linan M, Bowtell DD, Mes-Masson AM, Brenton JD, Aparicio SA, Boyd N, Hirst M, Gilks CB, Marra M, Huntsman DG. Mutation of FOXL2 in granulosa-cell tumors of the ovary. N Engl J Med. 2009 Jun 25;360(26):2719-29. doi: 10.1056/NEJMoa0902542). In this regard, changes in FOXL2, particularly hypermethylation, have been  reported recently in ovarian carcinomas especially in serous and mucinous histotypes and seem associated to a poorer prognosis (Losi L, Fonda S, Saponaro S, Chelbi ST, Lancellotti C, Gozzi G, Alberti L, Fabbiani L, Botticelli L, Benhattar J. Distinct DNA Methylation Profiles in Ovarian Tumors: Opportunities for Novel Biomarkers. Int J Mol Sci. 2018 May 24;19(6). pii: E1559. doi: 10.3390/ijms19061559). Therefore, interaction between FOXL2  alterations and TGF-β superfamily signaling pathway will help  understand the pathogenesis of ovarian tumors and design novel therapeutic strategies.”

Author Response

-          At page 2, I think it is  anti-Müllerian instead of Müellerian.

Correction made

-          In the third section of the review, TGF-β signaling in ovarian cancer, I think that some informations could be added about the role of TGF-β pathway and its related genes in ovarian tumor in general, considering their great heterogeneity based on molecular, morphological and clinical characteristics, although epithelial  tumors are  the most frequent.

Paragraph added (see below):

Ovarian cancer is classically divided into three different tumor types: epithelial, sex-cord stroma, and germ cell tumors, with over 90% classified as epithelial malignancies.  Within the epithelial tumors alone, there are five main histotypes: low grade and high grade serous, clear cell, endometrioid, and mucinous carcinomas.  On a histologic level, great herterogeneity exists which is further characterized on a molecular level.  Despite these well-documented differences, which greatly impact prognosis and clinical outcomes, similar treatment strategies and therapies are utilized across all ovarian cancer subtypes.  It is important to identify pathways with differential signaling that contribute to the development of these malignancies.  Existing literature supports the hypothesis that disruption of TGF-β superfamily genes result in development of human sex-cord stromal tumors, especially granulosa cell tumors (GCT). New evidence has revealed molecular and functional interactions between Smad3, a member of Smad complex involved in the intracellular signaling of the TGF-β canonical pathway, and FOXL2. This interaction regulates the activity of genes associated with cell proliferation and survival suggesting that Smad3 activation could serve as a critical converging point between dysregulated  TGF-β superfamily signaling and genetic alterations in human GCT development.  Indeed, FOXL2 is reported mutated in most   granulosa cell tumors. In this regard, changes in FOXL2, particularly hypermethylation, have been reported recently in ovarian carcinomas especially in serous and mucinous histotypes and seem associated to a poorer prognosis.  Therefore, interaction between FOXL2 alterations and TGF-β superfamily signaling pathway will help understand the pathogenesis of ovarian tumors and design novel therapeutic strategies especially in this particular histologic subtype. 

Reviewer 2 Report

The Review article Targeting the Transforming Growth Factor-Beta Pathway in Ovarian Cancer is very interesting. However, the article can be improved.

1.       The article needs to include more information about the targeting of the TGFbeta pathway in the context of immuno oncology. There are a number of studies recently reported that blocking of the check point inhibitors along with TGF beta antibodies as a potential therapy. The authors can add some detail about that.

2.       There is emerging evidence that the TGFbeta receptor I is cleaved like the Notch receptor. This could be a novel therapeutic target in various cancers. The authors could add this information in the article.

3.       The cross talk between various pathways would be useful to know about the dynamics of signaling in Ovarian cancer.

Author Response

The Review article Targeting the Transforming Growth Factor-Beta Pathway in Ovarian Cancer is very interesting. However, the article can be improved.

1.       The article needs to include more information about the targeting of the TGFbeta pathway in the context of immuno oncology. There are a number of studies recently reported that blocking of the check point inhibitors along with TGF beta antibodies as a potential therapy. The authors can add some detail about that.

Tauriello et al, designed a quadruple-mutant mouse model that spontaneously generates metastatic intestinal tumors.  Anti-PD-L1 therapy, yielded limited response in the model; however, inhibition of TGF-β with a small molecule inhibitor produced a robust cytotoxic T-cell response preventing metastasis and increasing susceptibility to anti-PD-L1 therapy [93].  The combination of anti-TGF-β therapy and immune checkpoint inhibition has also been studied in a preclinical breast cancer model.  In vivo treatment of mice with a TGF-β inhibitor resulted in almost 100% tumor growth inhibition with regression in about half the mice once therapy was discontinued.  In those mice that had a durable regression, upon re-challenge with 4T1-LP tumor cells, they remained tumor free suggesting the establishment of an immunological memory.   This same study combined anti-PD-L1 antibody with anti-TGF-β thearpy in a colon cancer mouse model which resulted in an improved response when compared to anti-PD-L1 alone [94].

A pair of studies described fusion proteins that simultaneously target TGF-β receptors and PD-L1 along with TGF-β receptors and CTLA-4.  This combined blockade strategy, resulted in marked reduction of tumor burden in pre-clinical models, further supporting the growing body of literature that supports the role of TGF-β inhibition in combination with immunomodulatory agents [96,97].  These combination strategies are not limited to immune checkpoint inhibitors as outlined above, but also includes OX40 antagonistic antibodies [98], IL-2 [99], radiation therapy [100], and oncolytic viruses [101].

Newsted et al, identified a lead inhibitory antigen-binding fragment (Fab) via screening of a synthetic library with TβRII.  Suppression with this lead Fab resulted in reversal of EMT and tumor invasion and an improved response to chemotherapy in EOC xenograft and syngeneic models [115]

2.       There is emerging evidence that the TGFbeta receptor I is cleaved like the Notch receptor. This could be a novel therapeutic target in various cancers. The authors could add this information in the article.

Smad-independent aberrations have also been identified as drivers of tumorgenesis.  There is evidence that within cancer cells, TβRI undergoes cleavage by tumor necrosis factor (TNF) alpha converting enzyme (TACE) resulting in the translocation of the receptor intracellular domain to the nucleus.  This leads to activation of the p38-mitogen activated kinase (MAPK) pathway which has been associated with EMT.

Further, ubiquitin-specific protease (USP) 4 has also been implicated in the cleavage of TβRI.  This cleavage activity of the transmembrane receptor via USP4 controls receptor levels along the cellular membrane.  Depletion of USP4 blocks TGF-β-induced EMT.  Stability of USP4 is dependent on AKT phosphorylation which leads to cytoplasmic migration of USP4 from the nucleus to the cytoplasm and cellular membrane where it can act on TβRI.  USP4 serves as a link between TGF-β and AKT signaling via cleavage of TβRI

3.       The cross talk between various pathways would be useful to know about the dynamics of signaling in Ovarian cancer.

Crosstalk between TGF-β and other well-known molecular pathways are being studied in ovarian cancer models as well.  Inhibition of TGF-β signaling via LY2109761, was shown to reduce tumor burden in orthotopic ovarian cancer PDX models. This inhibition resulted in a reduction in mRNA expression of insulin growth factor 1 (IGF-1) receptor.  The therapeutic effect of TGF-β inhibition is lost when IGF-1 receptor signaling is blocked.  These findings indicate that IGF-1 receptor signaling is positively regulated by TGF-β and is associated with ovarian tumor progression.     

Smad-independent aberrations have also been identified as drivers of tumorgenesis.  There is evidence that within cancer cells, TβRI undergoes cleavage by tumor necrosis factor (TNF) alpha converting enzyme (TACE) resulting in the translocation of the receptor intracellular domain to the nucleus.  This leads to activation of the p38-mitogen activated kinase (MAPK) pathway which has been associated with EMT.  Further, ubiquitin-specific protease (USP) 4 has also been implicated in the cleavage of TβRI.  This cleavage activity of the transmembrane receptor via USP4 controls receptor levels along the cellular membrane.  Depletion of USP4 blocks TGF-β-induced EMT.  Stability of USP4 is dependent on AKT phosphorylation which leads to cytoplasmic migration of USP4 from the nucleus to the cytoplasm and cellular membrane where it can act on TβRI.  USP4 serves as a link between TGF-β and AKT signaling via cleavage of TβRI.
